# Theoretical Background of Occupational-Exposure Models—Report of an Expert Workshop of the ISES Europe Working Group “Exposure Models”

**DOI:** 10.3390/ijerph19031234

**Published:** 2022-01-22

**Authors:** Urs Schlüter, Susan Arnold, Francesca Borghi, John Cherrie, Wouter Fransman, Henri Heussen, Michael Jayjock, Keld Alstrup Jensen, Joonas Koivisto, Dorothea Koppisch, Jessica Meyer, Andrea Spinazzè, Celia Tanarro, Steven Verpaele, Natalie von Goetz

**Affiliations:** 1Unit “Exposure Scenarios”, Federal Institute for Occupational Safety and Health—BAuA, Friedrich-Henkel-Weg 1-25, 44149 Dortmund, Germany; Meyer.Jessica@baua.bund.de; 2Division of Environmental Health Sciences, School of Public Health, University of Minnesota MMC 807, Room 1239 Mayo, 420 Delaware Street SE, Minneapolis, MN 55455, USA; arnol353@umn.edu; 3Department of Science and High Technology, University of Insubria, via Valleggio 11, 22100 Como, Italy; francesca.borghi@uninsubria.it (F.B.); andrea.spinazze@uninsubria.it (A.S.); 4Institute of Occupational Medicine, Research Avenue North, Edinburgh EH14 3AP, UK; john.cherrie@iom-world.org; 5Institute of Biological Chemistry, Biophysics and Bioengineering, Heriot-Watt University, Riccarton, Edinburgh EH14 3AS, UK; 6Department Risk Analysis for Products in Development, TNO, P.O. Box 80015, 3508 TA Utrecht, The Netherlands; wouter.fransman@tno.nl; 7Cosanta BV, Stationsplein Noord-Oost 202, 1117 CJ Schiphol-Oost, The Netherlands; Henri.Heussen@Cosanta.nl; 8Jayjock Associates, LLC, 168 Mill Pond Place, Langhorne, PA 19047, USA; mjayjock@gmail.com; 9National Research Centre for the Working Environment, 105 Lersø Parkallé, DK-2100 Copenhagen, Denmark; kaj@nfa.dk; 10ARCHE Consulting, Liefkensstraat 35D, 9032 Wondelgem, Belgium; joonas.koivisto@arche-consulting.be; 11Institute for Atmospheric and Earth System Research (INAR), University of Helsinki, PL 64, FI-00014 Helsinki, Finland; 12Air Pollution Management, Willemoesgade 16, st tv, DK-2100 Copenhagen, Denmark; 13Section Exposure Monitoring—MGU, IFA—Institute for Occupational Safety and Health of the German Social Accident Insurance, Alte Heerstrasse 111, 53757 Sankt Augustin, Germany; Dorothea.Koppisch@dguv.de; 14Exposure and Supply Chain Unit, European Chemicals Agency, ECHA, P.O. Box 400, FI-00121 Helsinki, Finland; celia.tanarro@echa.europa.eu; 15Health, Environment and Public Policy Department, Nickel Institute, Rue Belliard 12, 1040 Brussels, Belgium; sverpaele@nickelinstitute.org; 16Belgian Center for Occupational Hygiene (BeCOH), Technologiepark 122, 9052 Ghent, Belgium; 17FOPH—Federal Office of Public Health, Schwarzenburgstr. 157, 3003 Berne, Switzerland; natalie.vonGoetz@bag.admin.ch

**Keywords:** occupational-exposure modelling, mass-balance model, modifying-factor model, regulatory exposure modelling, workshop

## Abstract

On 20 October 2020, the Working Group “Exposure Models” of the Europe Regional Chapter of the International Society of Exposure Science (ISES Europe) organised an online workshop to discuss the theoretical background of models for the assessment of occupational exposure to chemicals. In this report, participants of the workshop with an active role before and during the workshop summarise the most relevant discussion points and conclusions of this well-attended workshop. ISES Europe has identified exposure modelling as one priority area for the strategic development of exposure science in Europe in the coming years. This specific workshop aimed to discuss the main challenges in developing, validating, and using occupational-exposure models for regulatory purposes. The theoretical background, application domain, and limitations of different modelling approaches were presented and discussed, focusing on empirical “modifying-factor” or “mass-balance-based” approaches. During the discussions, these approaches were compared and analysed. Possibilities to address the discussed challenges could be a validation study involving alternative modelling approaches. The wider discussion touched upon the close relationship between modelling and monitoring and the need for better linkage of the methods and the need for common monitoring databases that include data on model parameters.

## 1. Introduction

### 1.1. Background

To comply with the REACH (Registration, Evaluation, Authorization and Restriction of Chemicals) regulation (EC 1907/2006), quantitative occupational exposure assessments are needed to evaluate the occupational risk of chemicals for all the identified relevant exposure scenarios [1]. As defined by the European Chemical Agency (ECHA), it is possible to assess occupational exposure to chemicals by modelling, as an alternative or to complement workplace monitoring [1]. Beyond this specific regulatory context, the use of occupational-exposure models also finds application in risk-assessment approaches related to workplace-specific legislation. For example, the use of exposure modelling is also considered in a recent European standard [2] as part of the basic occupational risk assessment. This standard indicates that, in addition to exposure measurements, results from exposure modelling may be used as guiding values when appropriate models or algorithms are used. Therefore, a constant development of exposure models is necessary and has been taking place. The assessment workflow outlined by ECHA suggests a tiered approach to obtain quantitative exposure estimates: the use of simple-to-use and conservative exposure models is recommended in the first step of the evaluation (Tier 1), while the use of more complex and detailed models is recommended, when necessary, in subsequent assessments (Tier 2 and higher Tiers). The ECHA guidance mentions five exposure tools for assessment of occupational exposure by inhalation. As Tier 1, the following three models are most commonly used for REACH purposes: European Centre for Ecotoxicology and Toxicology of Chemicals Targeted Risk Assessment (ECETOC TRA), MEASE 2, and the EMKG Expo-Tool. The only suggested Tier 2 model is the Advanced REACH Tool (ART; www.advancedreachtool.com, accessed on 13 December 2021), whereas STOFFENMANAGER^®^ (www.stoffenmanager.com, accessed on 13 December 2021) is generally considered to be a tool between Tier 1 and Tier 2 [1]. Tier 1 and Tier 2 models in this context differ in particular in that a lower-tier exposure model should produce more conservative exposure estimates than a higher-tier exposure model [3]. The available exposure models and tools vary by domain of applicability, level of detail, and type of results. Additionally, there is also a different level of acceptance for some of the exposure models, depending, for example, on the evaluating authority or the legal framework. Outside the REACH framework, more and different models are being used, as described below. A brief description and appraisal of the models that are frequently used for REACH purposes has recently been provided [4].

### 1.2. Challenges in Occupational-Exposure Modelling

The Editorial “How Accurate and Reliable Are Exposure Models?” published in the *Annals of Work Exposures and Health* [5] provides a brief but comprehensive overview of the current status of regulatory occupational-exposure models. In that issue of the *Annals*, several authors report on the validation of Tier 1 exposure models used in the context of REACH ([6,7,8]). These papers show that the efforts of the occupational hygiene and exposure science communities to develop useful generic exposure-assessment models and tools succeeded to an extent that allows the use of these models as risk-assessment tools under REACH. The acceptance of these models for occupational safety and health purposes depends on the specific situation and the relevant national legislation. However, the results of the different evaluation exercises for these models show that their validation status should be further improved [4,9]. Therefore, modelling results need to be interpreted with caution and more knowledge is needed about different aspects of these models (e.g., model functionalities, applicability domains, uncertainties, between-user reliability) in order to use them in a meaningful way.

Differences in the interpretation of the theoretical background of two specific models (i.e., STOFFENMANAGER^®^ and ART) have led to controversy regarding the accuracy of some of the models‘ components and subsequent results [10,11,12,13,14]. To contribute to this ongoing scientific discussion, this workshop was organised to discuss the challenges in occupational exposure modelling and the theoretical background of some of the existing occupational-exposure models and how they can be improved. In the aftermath of the workshop, Koivisto et al. performed a theoretical evaluation of STOFFENMANAGER^®^ and ART [15].

### 1.3. Workshop Organization

ISES Europe (the European Regional Chapter of the International Society of Exposure Science) identified exposure modelling as one priority area for the strategic development of exposure science in Europe in the coming years [16,17]. Additionally, the ongoing scientific discussion about parts of the theoretical background and applicability of some frequently used tools/models in occupational-exposure assessment (i.e., ART and STOFFENMANAGER^®^; [10,11]) and mass-balance-based modelling approaches (e.g., [18] or [19]) in general highlights the importance of occupational-exposure models for regulatory purposes and their relevance for exposure science. For these reasons, the Working Group “Exposure Models” of ISES Europe [20] took the initiative to organise this workshop in order to have an open scientific exchange with the aim of discussing the challenges and opportunities as well as potential ways forward that are described below in more detail.

Participants discussed the applicability and the limitations of some of the currently available modelling approaches with the intention of reaching a common understanding of the benefits and limitations of the different approaches of modelling occupational exposure. For some challenges, an attempt was made to draw a roadmap towards future exposure-modelling initiatives. To accomplish this, the workshop was characterised by mutual trust and respect for the competence and work of all the scientists, regulators, industry representatives, and other participants.

#### 1.3.1. Topics under Discussion

During the Workshop, the challenges described in Table 1 were presented and discussed by the participants. A detailed timeline of the workshop, the slides of the presentations, and a background document are available on the ISES Europe website [20].

#### 1.3.2. Workshop Format and Participants

The Working Group organised the workshop as a virtual meeting, with the patronage of ISES Europe. It was open and free for anybody with an interest in exposure modelling and about 70 participants engaged in discussing the applicability of the currently available modelling approaches for workplace exposure assessment.

The agenda and scientific information that was distributed to the workshop participants in advance facilitated the discussions; the presentations given during the workshop gave a clear understanding of the various exposure-modelling approaches.

Scientists involved in the development of the models introduced different approaches for occupational exposure modelling and the theoretical background on which these approaches are based. In addition, some other participants were particularly active in the preparation of and the discussion during the workshop due to their substantial knowledge and experience with exposure modelling. These scholars attended the workshop to hear first-hand the debate and to contribute to the discussion and were also involved in the dissemination of workshop results. The workshop was organised and moderated by Natalie von Goetz and Urs Schlüter.

All participants acted in an individual capacity and not as representatives of any organization or committee to which they are currently or were previously associated.

After each presentation, all participants briefly discussed the respective topics in a plenary session with all experts and participants. After all presentations, experts and participants discussed all the topics in a plenary discussion collecting impressions, opinions, criticisms, and proposals. The Working Group had prepared the following key questions to structure this plenary discussion:How can the ISES Europe Working Group promote validation exercises for occupational-exposure models?How can the ISES Europe Working Group further develop occupational-exposure models for regulatory purpose?For what specific regulatory questions can different stakeholders use occupational-exposure models?What has to happen to improve occupational-exposure models in the future?

The most interesting aspects of the discussion were collected by the moderators, who, together with additional active participants, prepared this workshop report to present the collective views of all the experts.

## 2. Workshop Outcomes: Expert’s Comments on the Theoretical Background of Occupational-Exposure Models

### 2.1. Concept of Modifying-Factor Approaches (STOFFENMANAGER^®^ and ART)

STOFFENMANAGER^®^ and ART are based on a modelling approach published by Cherrie et al. in 1996 [21]. These tools apply this modelling approach by using a set of multipliers for the principal modifying factors (including the source) where the magnitude of the multiplier can be selected by the tool user based on the scenario, and from this calculating a score, which is then converted to a concentration using empirical calibration data from inhalation-exposure measurements. Multipliers for the modifying factors were derived based on data from literature, and chemical and physical laws. Where this was not possible, expert judgement was applied to derive multipliers for the assessment procedure. Quantification of multipliers is possible by using measurement data, both historical and from specially conducted surveys, e.g., [22,23], and contextual information. Multipliers for all modifying factors were peer reviewed by leading experts from industry, research institutes, and public authorities. In addition, experts discussed the proposed exposure multipliers in several workshops. Calibration equations were assigned using a linear mixed-effects model. In simple terms, the calibration equations thereby are determined by relating the measured exposure values to the modifying factors, which are in turn selected based on the contextual information collected for each measurement. The calibration equations are assigned for four different exposure groups [23,24]. In total, the tool developers used more than 2000 measurements for the calibration that they considered as good quality. Between-company, between-worker, and within-worker component of variance were introduced to reflect exposure variability and model uncertainty [11,23]. This methodology raises a number of questions discussed in the workshop background document [20] about the uncertainties in mass-balance or modifying-factor models and their predictive capacity [15,19]. This workshop tried to address and find a way to answer some of these questions, attempting to develop a common understanding of these issues.

STOFFENMANAGER^®^ and ART are well-established and accepted tools that are used for regulatory risk assessments, e.g., for REACH or in the context of the EU occupational health and safety directive 98/24/EC [11]. According to Cherrie et al. [11], another key strength of these models is that exposure can be estimated for loosely specified exposure scenarios, as the variations found at real workplaces are included in the modelling process.

Some workshop participants asserted that a limitation of these models is that they lack a physical basis. The developers explained that the models are based on a combination of physical laws (e.g., Raoult’s law) complemented by documentation from literature and expert judgement to fill knowledge gaps. By calibration of these models, the dimensionless multipliers are translated into quantitative exposure estimates in mg/m^3^, which serves to provide insight into the variability and uncertainty of the exposure model estimates and helps the user to select an appropriate level of conservatism. However, no consensus was reached at the workshop on whether this approach is described transparently enough in the scientific literature.

### 2.2. Mass-Balance-Based Exposure-Modelling Approaches

Mass-balance-based exposure-model approaches rely on physicochemical relationships and the conservation of mass. A range of models are available, from lower tier (screening level) to higher tier (refined estimates) [25]. The source strength of emission is one of the most important parameters in mass-balance-based exposure modelling. Mostly, measurements are necessary if chemical principles cannot be used to determine the source strength, especially when the source is complex (e.g., for aerosols). Unfortunately, such measurements are not yet widely available (databases for indoor air pollutant emissions that are publicly available are of low relevance for occupational exposure assessment), though the availability of real time monitoring instruments has made source characterization easier. If measurements of the source strength are not available, a tiered approach needs to be followed, including the estimation of emissions, e.g., by assuming that all process losses are emitted to air or read-across to comparable situations.

In principle, specific exposure situations can be modelled very accurately with mass-balance-based modelling approaches when a detailed scenario and contextual information exist [26]. Since these modelling approaches are based on fundamental physicochemical laws, balance principles, and measurable input parameters (e.g., source strength), underlying exposure-influencing processes may be analysed. In practice, the lack of reliable values for input parameters can be a problem in occupational settings, where usually a high within- and between-worker-variability can be found [27]. Another limitation is the lack of direct applicability of simple mass-balance modelling for general-scenario exposure assessments if not coupled to, e.g., range-values or known statistical variability of the determinant parameters for the specific scenarios assessed.

Within a tiered approach, different modelling approaches can be used for exposure assessments. Next to control-banding approaches and modifying-factor approaches, mass-balance based modelling can also be applied for occupational exposure assessment. All these approaches can also follow a tiered approach. Higher-tier mass-balance-based models provide a refined quantitative estimate of exposure, but require a nontrivial investment in quantifying the model inputs and are often complex, requiring a level of expertise beyond novice to identify or generate the required input data and appropriately interpret their output.

All models, including mass-balance-based modelling approaches, make simplified assumptions that have effects on the modelling result and introduce systematic uncertainties. Additionally, more sources of uncertainty are relevant for all models, such as the availability and quality of measured data and input parameters.

Currently, there are no generic mass-balance-based models that are described in the relevant European guidance (e.g., [1]) to assess exposure for regulatory risk assessments for workplaces.

### 2.3. Requirements for the Validation of Models

Different interpretations exist for the term “validation” of models. Tischer et al. [6] proposed one possible meaning: Validating a model means evaluating if: (i) the concept and theory of the model is sound or generally accepted; (ii) the output is accurate and precise compared to the “real” exposure; (iii) the tool in which the model is imbedded is user-friendly; and (iv) the between-user reliability is acceptable.

To evaluate the concept and theory of the model background, principles and methods used to derive the model equations are examined. This information should be well documented and publicly available. Additionally, the model developers need to describe clearly the applicability domain.

Whether a model output is accurate and precise can be answered by comparing the tool estimates with an independent set of measurement data (e.g., [28,29,30]) ideally covering a wide range of exposure scenarios and substances. For this exercise, good-quality measurement data are needed, for which all relevant variables (amongst others: workplace description, worker characterisation, sampling protocol, and analytical sensitivity related) are documented. Different statistical parameters are used such as correlation coefficients, bias, precision, ratios of modelled and measured exposures, and the percentage of measurements that exceed the percentile that is estimated. When choosing a specific set of statistical parameters, it is important to include parameters which consider the high variability of exposures in occupational settings.

User-friendliness of a tool is often evaluated in interviews, questionnaires, or workshops. In a structured way, methods of usability testing or usability inspection can be used.

The between-user reliability can be evaluated by asking a group of users to assess independently a set of the exposure scenarios by giving them the same description of workplaces for each of these exposure scenarios and then comparing the individual results [7].

Despite the extensive experience of exposure scientists in evaluating models and tools that has been gained in recent years, no standard describing the minimum requirements for model validation has yet been defined. There is also no independent institution designated to assess the validity or regulatory acceptance of exposure models.

### 2.4. Special Requirements for Regulatory Exposure Modelling

A number of tools (amongst others, STOFFENMANAGER^®^ and ART) are widely accepted in regulatory exposure assessment for different regulatory frameworks (e.g., REACH guidance documents). ECHA accepts some tools for exposure assessment in the framework of the REACH regulation or Biocidal Products Regulation. EFSA, on the other hand, recommends different models and tools for the assessment of occupational exposure to plant protection products [31]; that is, e.g., herbicides and insecticides for plant protection. In chemical risk assessment, an understanding of the model and the model inputs and outputs is needed to assess the reliability and applicability of the modelling results. Proper knowledge of exposure determinants (i.e., the model parametrization) and the transparent publication of these in the peer-reviewed literature is key to efficient and well-justified decision making.

Despite the important role of exposure modelling for decision-making, the criteria for models that are used for regulatory purposes are not well defined. There is, for example, currently no agreement on the level of conservatism (i.e., the tendency to overestimate the exposure, to provide a conservative assessment) or realism (i.e., the tendency to provide an accurate exposure estimation, to provide a realistic assessment) necessary for the different fields of regulatory decision making.

## 3. Discussion

### 3.1. Theoretical Background and Validation of Models

In the workshop discussions, the developers explained the theoretical background, differences, and validation status of the widely used tools STOFFENMANAGER^®^ and ART as follows: ART is a multiplicative model based on modifying factors identified from a theoretical analysis and calibration. STOFFENMANAGER^®^ is based on a simpler set of equations than ART, although the two models share a similar concept. Some of the participants considered the foundations of these models as well described and supported with sound scientific evidence in the peer reviewed literature. However, some of the participants argued that further evidence and explanations should be provided about the derivation of model multipliers and of the calibration factors, and about the physical concept of these tools, which, according to them, was not described clearly enough by tool developers [15]. Overall, no consensus was reached at the workshop on this issue, so this topic should be further and extensively addressed in the framework of future independent research studies. Mass-balance-based models follow a different approach and are more directly based on physical–chemical laws and parameters, source strength measurements, and scenario and contextual information than modifying-factor approaches.

For ART, all the information that was used for the development of the tool has been published in the peer-reviewed public domain. This includes, e.g., theoretical background, physical laws, harvested literature, expert judgement procedures, emission rates, and a description of the data that were used for the calibration of the model [32]: www.advancedreachtool.com/science.aspx/ (accessed on 13 December 2021). For STOFFENMANAGER^®^, a similar level of information is freely available on the model’s website [33]: https://stoffenmanager.com/what-is-stoffenmanager/ (accessed on 13 December 2021). The description of the data used for calibration has been published in the peer-reviewed literature. However, this information does not include the databases of measurements as such that were used in the calibration exercises.

The workshop participants agreed that testing of mass-balance-based approaches against measured exposure data from databases that were already used for tool-validation exercises would be a worthwhile activity. However, the effort for setting up this kind of study could be enormous (already for the very first phase of the study, i.e., data gathering, selection, and evaluation) and would need funding, expertise, and time. A second idea would be to initiate comparative studies for the different modelling approaches [34] based on the information that is usually available to regulatory exposure assessors. This would show more clearly how the different approaches relate to each other and could promote the advancement of the tiered approach of regulatory exposure assessment. Modelling approaches and measurements (be it workplace monitoring, task-specific measurements, or the quantification of model parameters) are both used in this tiered approach. The ART model has a built-in Bayesian module, which facilitates the combination of the model output with measurement data to derive a more accurate estimate of exposure [35]. However, a structured procedure is missing to integrate different modelling approaches and measurements within a single-paradigm tiered approach.

The workshop participants considered that some kind of platform with a common database is necessary to integrate information from the occupational health and safety framework and the chemical regulations as a prerequisite for the use of both modelling approaches. Furthermore, it needs to be defined beforehand which standard scenarios should be investigated as they should (i) have regulatory relevance, and (ii) allow compliance with applicability domains of tools and models, which are under investigation.

Some workshop participants with a regulatory background made clear that the validation status of models or tools is of high relevance for regulators and the quality of regulatory decisions. However, regulators have to rely on experts, as the regulatory bodies usually do not have resources (e.g., measurement data for comparison), nor the necessary expertise to evaluate the validity of models or tools. Preparation of a standard describing the minimum requirements for model validation and the establishment of an independent institution designated to assess the validity and regulatory acceptance of exposure models should be investigated.

### 3.2. More and Improved Measurements as Background for Improved Models

It is acknowledged and it was confirmed in the workshop discussion (see Figure 1) that models cannot and should not replace the collection of good-quality exposure measurements, where measurements are feasible. It is also the consensus view that models are often the best way to gain a general understanding of specific processes and to make prospective or retrospective assessments. In any case, for both modelling approaches (modifying-factor and mass-balance-based exposure-modelling approaches), exposure measurements as a background of the model are necessary. For modifying-factor models, exposure measurements are useful to decide on parameters that drive the exposure and define multipliers for the principle modifying factors; workplace-monitoring data are necessary to calibrate the model results against realistic workplace exposure situations.

Similar to that, for reliable mass-balance-based exposure-modelling approaches, measured values for model input parameters (in particular, emission rates) are required. In the absence of measurements, input parameters can be estimated, but with the acknowledgement that this introduces uncertainty into the model output. In addition, workplace monitoring is used to evaluate model approaches and model performance.

It also needs to be kept in mind that workplace exposure measurements have variabilities and uncertainties, which apply for both modelling approaches.

In the course of the workshop, the question came up: If measurements are necessary anyway for both modelling approaches, what is the reasoning for performing modelling in general and mass-balance modelling in particular? Would it not be more reasonable to invest in workplace monitoring?

To answer of these questions, the following different aspects need to be taken into account.

Measurements of parameters can be used more widely than workplace monitoring, as workplace monitoring is usually limited to specific situations and very often involves only a few samples.Parameters that are not available by measurements (e.g., emission values, etc.) can also be modelled, but of course are then subject to a higher degree of uncertainty.Some parameters (e.g., some physical data, etc.) are easily available in reference tables.Source emission rates are important values for modelling but need to be understood better. It would be beneficial to establish emission-rate libraries based on standard test methods.More measurement data (workplace monitoring, model parameters) may be available, but not in the public domain. The quality and accessibility of this data, however, are unknown and differ depending on the involved stakeholders. It is recognised that routinely collected exposure monitoring data often do not include the contextual parameters necessary for exposure modelling, e.g., effectiveness of local ventilation used or the behaviour of workers.Mass-balance-based exposure-modelling approaches may help to understand the underlying physical–chemical processes and thereby to identify the exposure-driving factors.Some legal frameworks require a risk assessment prior to the commencement of the activity. In this situation, monitoring is not possible and modelling or the use of existing exposure measurement data is the only option. Another example is workplace design where exposure modelling may be part of a Safe (and Sustainable)-by-Design strategy.

The workshop agreed that, for meaningful modelling, regardless of the chosen approach, reliable information must be available for defining the scenario. That means that some measurements of modelling input parameters or relevant exposure levels are always necessary. The quality of measured data and the related documentation are key elements. A common format for contextual information of workplace exposure monitoring would be an important step forward to acquire minimal information necessary for evaluating and improving exposure models. A discussion of these data needs has been previously published (e.g., [36] or [37]).

For the moment, it would be a significant step forward if all available information, different modelling approaches, and measurements could be properly taken into account for regulatory decisions. That means that a combination of modelling and measurements could be a solution [38]. Additionally, read-across of exposure measurement data may be considered a novel option to provide reliable exposure estimates for regulatory purposes [39]. This will make optimal use of existing exposure measurement data to be used for similar scenarios.

### 3.3. Subjectivity in Modelling Approaches and between-User Variability of Modelling Results

Between-user variability in the use of models was identified as relevant for modelling by different authors in the past (e.g., [7,40]). In addition, different interpretation of the model parameters by the users of models and tools can lead to different model outputs for the same exposure scenario. Appropriate training for model users is essential to reach acceptable comparability between users. Preferably, tool owners should strive to do more, e.g., develop and maintain an active user community, support of users with an interactive helpdesk, and regular options to discuss differences in model estimates. The workshop discussed that subjectivity and interpretation of model parameters by model users is relevant for both modelling approaches.

During model development, subjectivity shall be eliminated as much as possible, but may have an important influence, as the derivation of exposure modifying factors or simplifications to model structures require, to some extent, expert judgement.

### 3.4. Practicalities of Modelling in Regulatory Contexts

Participants of the workshop explained that developing mass-balance-based exposure-modelling approaches for thousands of scenarios for chemical regulations (e.g., REACH) is currently not possible for practical reasons. Both modifying-factor approaches and mass-balance-based exposure models have been applied for decades in regulatory occupational (e.g., [41,42]) and consumer (e.g., [43,44,45]) chemical safety assessments. At this time, it is also not possible to monitor a sufficient number of workplaces to account for the variance in exposure between workplaces. To address these limitations, the use of models is an option, at least for the screening of scenarios and substances and as a tool for risk management in individual companies.

For the moment, both modelling approaches are necessary and need to be used following the principles of a tiered approach. Some participants pointed out that mass-balance-based exposure-modelling approaches can follow a tiered approach, reducing model uncertainty as model complexity increases [46]. Exposure-modelling tools (modifying-factor approaches) are usually developed to address a specific level of accuracy in a tiered approach. Where modelling results show insufficient safety or where modelling is not possible, source measurements and workplace monitoring are needed to complete the exposure assessment. The ART model already has a built-in Bayesian module, which facilitates the combination of the ART-output with workplace monitoring data to derive a more accurate estimate of exposure [35]. The workshop participants concluded that both modelling approaches have their strengths and their weaknesses. For both approaches, continuous developments are necessary. Exposure scientists need to examine how they can integrate new technologies and methods into the modelling approaches (e.g., read-across for exposure measurements [39]). Additionally, other materials (e.g., fibres, nano-sized materials, mixtures) or exposure pathways (e.g., dermal, ingestion) need to be included in the applicability domains of the modelling approaches, either in modifying-factor or mass-balance-based approaches.

## 4. Conclusions

The workshop on the theoretical background of occupational-exposure models organised by the ISES Europe Working Group “Exposure Models”, aimed, among the objectives, to describe basic features of mass-balance-based and modifying-factor modelling approaches, identifying advantages and disadvantages of both approaches and classifying the reliability of the results. The general aim of the workshop was to discuss different opinions in a shared scientific context: this might help and support any further discussion that will arise in the future and base such discussion on a common ground. Highlights of the extensive and thorough experts’ discussion have been presented in this paper. However, the discussion (see, for example, [10,11,15,20]) is ongoing and requires an independent institution designated to assess the validity and regulatory acceptance of exposure-modelling tools.

Overall, after what has been reported above, there is still an unanswered question of whether one modelling approach is superior to other approaches. The superior model may be the one that can provide the best estimate, given the nature of the source, environment, and what is known about them. Using both types of models to triangulate an exposure estimate in many cases might prove to be better than either model type on its own. At the described expert workshop, the participants exchanged ideas on the theoretical background of occupational-exposure models, and identified challenges and ways forward, but this ultimately cannot solve problems or decide on solutions.

Results from an expert workshop can only be considered as one puzzle piece in the scientific discussion regarding appropriate modelling approaches. The workshop identified next steps that need to be initiated.

Although the goal of the workshop was to trigger activities directed at the improvement of the discussed occupational-exposure models, it is not yet clear under which framework the necessary validation and comparison efforts can be funded and accomplished. It is obvious that the promise of efficient exposure assessment through modelling will not be met without proper support.

The workshop participants identified a number of necessary actions for modelling approaches (be it mass-balance-based or modifying-factor approaches), which could improve estimation of occupational exposure for regulatory purposes. As a first step for common validation activities, data-exchange templates for exposure measurements in order to arrive at a common minimum set of variables and quality criteria would be beneficial. Only then, a common framework for the evaluation of models and tools comprising the models themselves, the model documentation, and user-friendliness of the tools is possible. An interesting research project could be the comparison of the tools in question to suitable mass-balance approaches. However, good parameterization is needed, which could be achieved by the above-mentioned data-exchange templates.

The workshop participants agreed that further model development and improvement requires international collaboration to meet global needs of chemical safety assessment. Since in Europe the chemical safety assessment and the improvement of exposure descriptions (modelling and measurements) are the responsibility of the manufacturer or importer of chemicals, funding by industry should be explored. The workshop participants generally observed that more systematic funding for exposure science is missing, but would be needed, and that this is especially the case for exposure modelling. If exposure science is funded, usually monitoring programmes are in the focus of funding bodies.

Ideally, a common platform for hosting and funding of models should be developed on a global level. The private (industry, business associations) and the public (funding agencies, regulatory bodies) sectors may both play a role in this effort. It should be noted that funding opportunities will not be emerging as long as the current state of the art of exposure assessment via modelling is deemed to be acceptable.

However, it is the conclusion of this workshop that exposure modelling in the service of regulatory decisions should be improved in many aspects. The ISES Europe Working Group “Exposure Models” is asked to look deeper into funding possibilities and promote exposure science and modelling with different stakeholders and funding agencies.

## Figures and Tables

**Figure 1 ijerph-19-01234-f001:**
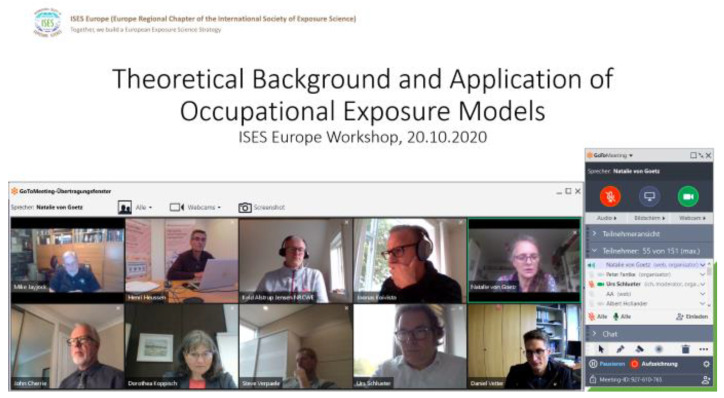
Discussion in the workshop.

**Table 1 ijerph-19-01234-t001:** Topics under discussion during the workshop.

Topic under Discussion	Presenter/Moderator (Affiliation)
Opening, logistics, agenda, aim of the workshop, moderation	Natalie von Goetz, BAG, Swiss Federal Office of Public Health Urs Schlüter, BAuA, Federal Institute for Occupational Safety and Health
Concept of STOFFENMANAGER^®^ and ART	John Cherrie, Heriot Watt University
Mass-balance modelling approach	Susan Arnold, University of Minnesota Joonas Koivisto, ARCHE Consulting
Requirements for the validation of models	Dorothea Koppisch, IFA, Institute for Occupational Safety and Health of the German Social Accident Insurance
Requirements for regulatory exposure modelling	Celia Tanarro, ECHA, European Chemicals Agency

## Data Availability

Not applicable.

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
