# Peer review of "Theoretical Background of Occupational-Exposure Models—Report of an Expert Workshop of the ISES Europe Working Group “Exposure Models”"

_ijerph, 2022, doi:10.3390/ijerph19031234_

Round 1
Reviewer 1 Report
This is a report of a recent Working Group meeting on exposure models. This will be of interest to people who work in occupational exposure and occupational epidemiology. I only have a couple of very minor comments
line 129: reference?
I dont think Figure 1 is necessary
Lines 191 -193: 'This methodology raises a number of questions. This workshop tried to address and find a way to answer some of these questions.' It wasnt quite clear what questions were raised and addressed. It would help if these were specifically stated
Author Response
The authors are very grateful for the precious suggestions and comments provided by the reviewers and for their feedback. They are greatly appreciated. The authors are now submitting a new version of the manuscript, that has been revised based on the reviewers’ concerns. The changes to the original manuscript were highlighted within the attached documents (track changes). A detailed, point-by-point list of replies to editor and reviewers’ comments was prepared and is included here.
1) line 129: reference?
Thank you for reporting; the typo was corrected as follows:
“During the Workshop, the challenges described in Table 1. were presented and discussed by the participants”
2) I don’t think Figure 1 is necessary
Thanks a lot for the suggestion. We kept the figure for the moment because the figure might serve as a graphic break that improves the flow of reading. We would leave it to the editors to make a decision. The illustration is not essential for understanding the text.
3) Lines 191 -193: 'This methodology raises a number of questions. This workshop tried to address and find a way to answer some of these questions.' It wasn’t quite clear what questions were raised and addressed. It would help if these were specifically stated
Thank you very much for reporting. The sentence was updated accordingly:
“This methodology raises a number of questions about the uncertainties in mass-balance or modifying-factor models and their predictive capacity [19]. This workshop tried to address and find a way to answer some of these questions.”.
Reviewer 2 Report
This is a very interesting communication about the results of a workshop of occupational exposure modeling: it is well written and interesting for the audience of the journal. I think it is fine to publish as is excepp by to very minor points.
1) in lines 77 and 291 the authors refers to being "conservative or realist". They could briefly explain what would be each in this context.
2) In line 128 it seem there is an error in a reference.
Author Response
The authors are very grateful for the precious suggestions and comments provided by the reviewers and for their feedback. They are greatly appreciated. The authors are now submitting a new version of the manuscript, that has been revised based on the reviewers’ concerns. The changes to the original manuscript were highlighted within the attached documents (track changes). A detailed, point-by-point list of replies to editor and reviewers’ comments was prepared and is included here.
1) in lines 77 and 291 the authors refer to being "conservative or realist". They could briefly explain what would be each in this context.
Thank you very much for the suggestion. The manuscript was updated accordingly in lines 292 – 294:
“There is, for example, currently no agreement on the level of conservatism (i.e. the tenden-cy to overestimate the exposure, to provide a conservative assessment) or realism (i.e. the tendency to provide an accurate exposure estimation, to provide a realistic assessment) necessary for the different fields of regulatory decision making.”
2) In line 128 it seem there is an error in a reference.
Thank you for reporting; the typo was corrected as follows:
“During the Workshop, the challenges described in Table 1. were presented and discussed by the participants”
Reviewer 3 Report
I strongly appreciate the efforts of the experts to solve the very current issues that they raised during the workshop.
Although I agree with the idea that discussion about the applicability and the limitations of some of the currently available modeling approaches to reach a common understanding of the benefits and limitations of the different approaches to model occupational exposure is needed, the benefits and novels presented in the submitted article are very poor. According to my opinion, the design of the article is very confusing for readers and, over more, does not bring a clear scientific message. In my opinion, with all humility and respect for the authors, this article is suitable for publication in a newsletter rather than in a scientific journal.
Author Response
The authors are very grateful for the precious suggestions and comments provided by the reviewers and for their feedback. They are greatly appreciated. The authors are now submitting a new version of the manuscript, that has been revised based on the reviewers’ concerns. The changes to the original manuscript were highlighted within the attached documents (track changes). A detailed, point-by-point list of replies to editor and reviewers’ comments was prepared and is included here.
We acknowledge this point. In this regard, we also realize that this manuscript does not present an innovative contribution to research, and in fact, we propose it as a "communication" rather than a pure scientific publication. The aim is to encourage the discussions of a topic of importance to occupational hygienists and exposure scientists. The aim is not to present new scientific information.
It is our expectation and hope that this communication may provide a new perspective on several issues related to occupational exposure modeling, but at the same time keeping the discussion based on evidence. Therefore, the authors did their best to illustrate the background, rationale and existing knowledge that led to the organization of the workshop, and then to present the main outcomes from the discussion between the experts at the workshop. We still think that it is important to submit this manuscript in order to document the discussion adequately that took place during the workshop. However, we also acknowledge the different currents of thought that emerged on the topics.
The aim of the workshop was to put in context these different views in a shared scientific discussion. In our opinion, this manuscript might help and support any further discussion that certainly will arise in the future and base such discussion on a common ground. We understand that a communication like this cannot be very exhaustive of a long complex and controversial discussion. Nevertheless, please keep in mind that the objective of this “workshop report” is to describe the presence of a divergence, rather than going into the detail of each of the issues under discussion, since an extended discussion of all details of this issue should not/cannot find space in this manuscript. Therefore, the goal is to point out that there is no unanimity on certain points, rather than exploring the details of the open questions.
Round 2
Reviewer 3 Report
In no way did my previous comments intend to disparage the work of the author's team. On the contrary, I appreciate the efforts of experts to solve this problem during a professional scientific conference by way of a theoretical approach. I understand that a special issue of your magazine is dedicated to just such article. Therefore I have reconsidered my decision, but I still have one substantial comment.
The final decision to publish this version is on the editor, but if it is possible, I still recommend using a graphical or table presentation of the strengths and weaknesses of the evaluated models. It would be an added value that would summarize the output of the diversity of opinion and at the same time the consensus resulting from the discussion of this conference.
Author Response
Thank you very much for your comments. We acknowledge this point. In this regard, very humbly we think that this workshop report could be of some value for the occupational hygiene / exposure science community (and we thank all reviewers for their positive feedback in this regard). However, we are unfortunately unable to fully address your last request regarding “a graphical or table presentation of the strengths and weaknesses of the evaluated models”. At the workshop the participating experts did not come to a consensus regarding strengths and weaknesses of the evaluated models. However, the aim of the workshop was to put in context different views in a shared scientific discussion. In our opinion, this manuscript might help and support any further discussion that will certainly arise in the future and base such discussion on a common ground. We understand our manuscript cannot be exhaustive of a long complex and controversial discussion. Nevertheless, please keep in mind that the objective of this “workshop report” is to describe the presence of a divergence, rather than going into the detail of each of the issues under discussion, since an extended discussion of all details of this issue should not/cannot find space in this manuscript
We updated the conclusion section of our manuscript to highlight these aspects:
“The workshop on the theoretical background of occupational exposure models organized by the ISES Europe Working Group “Exposure Models”, aimed, among the objectives, to describe basic features of mass-balance based and modifying-factor modelling approaches, identifying advantages and disadvantages of both approaches and classify the reliability of the results. The general aim of the workshop was to discuss different opinions in a shared scientific context: this might help and support any further discussion that will arise in the future and base such discussion on a common ground. Highlights of the extensive and thorough experts’ discussion have been presented in this paper. However, the discussion (see for example [10] [11], [15] and [20]) is ongoing and requires an independent institution designated to assess the validity and regulatory acceptance of exposure models. Overall, for what has been reported in the previous paragraphs, it is still an un-answered question if one modelling approach is superior to other approaches. The superior model may be the one that can provide the best estimate, given the nature of the source, environment, and what is known about them. Using both types of models to triangulate an exposure estimate in many cases might prove to be better than either model type on their own”.